# The Injected Foaming Study of Polypropylene/Multiwall Carbon Nanotube Composite with In Situ Fibrillation Reinforcement

**DOI:** 10.3390/polym14245411

**Published:** 2022-12-10

**Authors:** Gang Li, Yanpei Fei, Tairong Kuang, Tong Liu, Mingqiang Zhong, Yanbiao Li, Jing Jiang, Lih-Sheng Turng, Feng Chen

**Affiliations:** 1College of Material Science and Engineering, Zhejiang University of Technology, Hangzhou 310014, China; 2College of Mechanical Engineering, Zhejiang University of Technology, Hangzhou 310014, China; 3National Center for International Research of Micro-Nano Molding Technology, School of Mechanics and Safety Engineering, Zhengzhou University, Zhengzhou 450001, China; 4Department of Mechanical Engineering, University of Wisconsin–Madison, Madison, WI 53706, USA; 5Wisconsin Institute for Discovery, University of Wisconsin-Madison, Madison, WI 53715, USA

**Keywords:** PTFE, PP, MWCNTs, in situ fibrillation, injection foaming

## Abstract

This paper explored the injection foaming process of in situ fibrillation reinforced polypropylene composites. Using polypropylene (PP) as the continuous phase, polytetrafluoroethylene (PTFE) as the dispersed phase, multi–wall carbon nanotubes (MWCNTs) as the conductive filler, and PP grafted with maleic anhydride (PP–g–MA) as the compatibilizer, a MWCNTs/PP–g–MA masterbatch was prepared by using a solution blending method. Then, a lightweight, conductive PP/PTFE/MWCNTs composite foam was prepared by means of extruder granulation and supercritical nitrogen (ScN_2_) injection foaming. The composite foams were studied in terms of rheology, morphological, foaming behavior and mechanical properties. The results proved that the in situ fibrillation of PTFE can have a remarkable effect on melt strength and viscoelasticity, thus improving the foaming performance; we found that PP/3% PTFE showed excellent performance. Meanwhile, the addition of MWCNTs endows the material with conductive properties, and the conductivity reached was 2.73 × 10^−5^ S/m with the addition of 0.2 wt% MWCNTs. This study’s findings are expected to be applied in the lightweight, antistatic and high–performance automotive industry.

## 1. Introduction

In recent years, with the development of the economy, the preparation of technology that combines lightweight and high–performance polymer materials has become an important research direction for new energy vehicle materials [1,2,3,4,5,6,7,8]. PP has the largest market share in the automotive plastics market, accounting for approximately 21%. This is because PP has the smallest density (0.90~0.91 g/cm^3^) of all resins, with the density being only about 10% of steel. It is cost–effective, easy to form and recycle, excellent for heat resistance, chemical corrosion and stress cracking resistance [9]. However, traditional PP has poor flame–retardant performance, low–temperature performance and impact resistance, so now most of the automotive plastic PP is modified PP. PP–based microfibrillated composite materials have the advantages of light weight and high strength. The use of PP–based microfibrillated composite materials supplemented with microcellular foam molding can make a breakthrough in the production of lightweight automobiles.

Different from other polymers, PTFE possesses many desirable properties, such as excellent solvent resistance, high melting point and low yielding strength. The presence of fluoroalkyl functional groups in PTFE demonstrate strong thermodynamic affinity for CO_2_ [10,11,12]. The fibrillar network, which is developed by physical entanglements without dispersion difficulty for the fibrils with high aspect ratios, is formed from already well–dispersed phases [13]. Furthermore, PTFE particles can be fibrillated easily under shear extrusion due to its low inter–facial shear strength [14,15]. Hence, this paper chose PTFE as dispersed phase.

The in situ fibrillation mechanism of PTFE in PP is that as a dispersed phase, it will be deformed and oriented in the continuous phase PP due to shearing and stretching, and thereby form fibers. Park et al. [16] prepared in situ polymer–fibrillar blends of PP/PTFE and they found that adding only 0.3 wt% of PTFE is sufficient to markedly enhance the CO_2_ sorption capacity of the matrix, thereby improving PP foaming performance. In order to improve the thermal insulation performance of PP, Wang et al. [17] fabricated a thermal conductivity of as low as 36.5 mW·m^−1^·K^−1^ by high pressure foam injection molding, followed by mold–opening with CO_2_ as a blowing agent. Their study confirmed that PTFE fibers are very effective for improving melt strength and, thus, the foaming ability of PP. Subsequently, Park and Wang et al. have conducted a series of studies [18,19,20,21,22,23,24,25,26] to explore the mechanism of in situ fibrillation and its influence on foaming. Xie et al. [27,28] reported that the in situ fibrillated phase improved the filler’s dispersion and enhanced the reinforcing effect of the filler phase. Jurczuk et al. reported [15] that PTFE nanofibers nucleated the cells’ formation in long–chain branched (LCB–PP) and participated in controlling the cells growth. Recently, in situ fibrillation of different matrices has also been studied, such as PC [29], PLA [30] and so on. However, its application is limited due to its single performance; therefore, there is an urgent need to increase its field of application.

In order to solve the above problem, this paper uses PP as the continuous phase, PTFE as the dispersed phase, MWCNTs as the conductive filler, and PP–g–MA as a compatibilizer. The PP–g–MA–MWCNTs masterbatch is prepared by solution blending, granulating with a twin–screw extruder, and ScN_2_ injection foaming is used to prepare a lightweight, conductive composite foam. Different from the previous literature, this work adopted the MWCNTs selectively dispersed in the phase interface, and a low percolation threshold was achieved. The morphology, viscosity, and mechanical properties of composites and composite foams with different PTFE contents were tested. The optimal injection molding and foaming processing parameters were explored, and the conductivity of composite foams with different MWCNTs content were tested.

## 2. Experimental

### 2.1. Materials

Grade Pro–fax SG702 PP was purchased from LyondellBasell Industries (Rotterdam, Netherlands). PTFE was purchased from Daikin Industry Co., Ltd. (Osaka, Japan). Multi–walled carbon nanotubes (MWCNTs) were purchased from Aladdin Company (Shanghai, China). The nanotubes were >90% pure, with inner diameter of 5–15 nm, outer diameter of 30–80 nm and <10 μm long. Polypropylene grafted maleic anhydride (0.934 g/mL, maleic anhydride 8–10 wt%) was purchased from Aldrich Chemical Co., Inc with Mw~9100 (Milwaukee, WI, USA).

### 2.2. The Preparation of PP/PTFE Composites

In order to study the effect of different PTFE contents on the in situ fibrillation of PTFE/PP, PP/PTFE composites with PTFE contents of 1%, 3%, and 5% were prepared. PP and PTFE were weighed according to the mass ratio, then put in a beaker and stirred to ensure uniform dispersion, and then added to the feed port of the extruder (ZSE–18HP–e, Leistritz, Germany) for extrusion to obtain a PP/PTFE composite. The feed rate was 80 r/min and the speed was 150 r/min. The temperature of each zone of the extruder is shown in Table 1. 

### 2.3. The Preparation of PP–g–MA–MWCNTs Composite

A total of 10 g of MWCNTs was added to 1 L of DMF and then put into an ultrasonic cleaner with a power of 300 W to sonicate for 10 min to obtain a MWCNTs dispersion. Then 19 g of PP–g–MA was taken and dissolved in 100 g of p–xylene. Blends were added to the MWCNTs dispersion when they were dissolved, and a magnetic stirrer with the rotating speed of 1200 r/min was used for 3 h at room temperature to form a uniform dispersed mixture. The dispersion was poured into 1 L of ice methanol, the supernatant removed after the black solid was separated out, and a vacuum filtration was performed. Vacuum drying, at 60 °C for 12 h to remove residual DMF and methanol to obtain the sample, was then performed. The synthetic route is shown in Figure 1.

### 2.4. Masterbatch Preparation

The sample obtained above was added to 171 g of PP, after stirring and dispersion, and was then added to the twin–screw extruder for extrusion granulation to obtain a masterbatch with a MWCNTs content of 5%. The specific formula is shown in Table 2. The equipment and processing conditions are consistent with Section 2.2.

### 2.5. The Preparation of PP/PTFE/MWCNTs Composite

The PP, PTFE and masterbatch were blended according to the formula in Table 3, and PP/PTFE, PP/PTFE/0.2% MWCNTs, PP/PTFE/0.5% MWCNTs, and PP/PTFE/1% MWCNTs composites were prepared using a twin–screw extruder, and the final granulation was used for ScN_2_ injection foaming.

### 2.6. PP/PTFE/MWCNTs Composite Injection Foaming

The pellets obtained above were formed under the injection foaming process (Allrounder 320S, Arburg, Germany). The conditions were as follows: the melt temperature was 180 °C, the mold temperature was 50 °C, the filling was 12 cm^3^, the injection speed was 50 cm^3^/s, the N_2_ concentration was 0.4 wt%, the holding pressure was 15 MPa, the holding time was 40 s, and the cooling time was 30 s.

### 2.7. Morphological Characterization

#### 2.7.1. TEM

After ultra–thin sectioning of the PP/PTFE/MWCNTs composite foams, the JEM-1230 transmission electron microscope (TEM, JEM–1230, JEOL Ltd., Tokyo, Japan) was used to observe the dispersion of PTFE and MWCNTs in the composites under an acceleration voltage of 200 kV.

#### 2.7.2. SEM

Scanning electron microscopy (SEM, Vega 3, Tescan, Brno, Czech Republic) was used to observe the cross–sectional morphology of PP/PTFE/MWCNTs composites and their relevant foamed composites. The samples were placed in liquid nitrogen for brittle fracture treatment. At the same time, in order to observe the fibrillation of PTFE, the fracture surface of the PP/PTFE composite was steam etched with p–xylene under the condition of 65 °C, for 2 h. Then, an ion sputtering instrument was used to spray platinum on the surface of the section to enhance conductivity and observe the morphology of the sample with a Tescan Vega 3 desktop scanning electron microscope; the voltage was 10 kV.

For the SEM images, the analysis software Digital Micrograph was used to calculate the average cell size and cell density of the sample, and the Formula (1) was used to calculate the cell density *N* (units/cm^3^).
(1)$$N={\left[\frac{\left(nM^{2}\right)}{A}\right]}^{\frac{3}{2}}$$
where n is the number of cells in the SEM image, *M* is the magnification, and *A* is the area of the selected SEM area.

### 2.8. DSC Analysis

DSC (Q2000, TA, DE, USA) was used to calculated the thermal properties of PP/PTFE/MWCNTs composites. A 5–10 mg sample was weighed and placed in a sealed aluminum crucible. During the test, N_2_ was continuously introduced as a protection, and the gas flow rate was 50 mL/min. The samples were increased from room temperature to 200 °C at a temperature increase rate of 10 °C/min to remove thermal history. The samples were kept at a constant temperature of 200 °C for 3 min, then they were cooled down to room temperature at a cooling rate of 10 °C/min, and finally reheated to 200 °C for the second time.

### 2.9. Rheology Test

The rotary rheometer (AR2000, TA, DE, USA) was used to measure the rheological properties of the sample under shear flow. Samples were hot pressed into a film with a diameter of 25 mm and a thickness of 0.5 mm. The parameter strain value was set to 1%, the frequency swept from high frequency to low frequency, the angular frequency range was 0.03–200 rad/s, and the frequency sweep temperature was set to 180 °C.

### 2.10. Density Test

The drainage method was used to measure the density of the samples, the experiment was carried out at room temperature, and the density bottle was *V*_0_ = 25 mL. First, the mass *m*_0_ of the density bottle was measured, and then the mass *m*_1_ of the density bottle filled with water was measured. The mass of the sample as *m_s_* was measured, then the sample was slowly stuffed into the density bottle, the density bottle filled with water and closed, and the mass *m*_2_ of the sample and the density bottle lastly measured. The density *ρ* of the composite material was then calculated using Formula (2).
(2)$$\rho =\frac{m_{s}\left(m_{1}-m_{0}\right)}{V\left(m_{s}+m_{1}-m_{2}\right)}$$

### 2.11. Tensile Stress

The tensile test adopts the standard tensile dumbbell–type samples. According to ASTMD 638, the tensile speed was 50 mm/min at a temperature of 25 ± 2 °C, and each group of samples was tested at least 5 times, and the average value was finally taken.

### 2.12. Conductivity Test

A rectangular spline was prepared with length *L* = 8 mm, width *W* = 2 mm, and thickness *D* = 2 mm; the positive and negative poles of the electrochemical workstation (RST5000) were connected with copper wires at both ends of the spline, and the single–potential step chronoamperometry was selected. The measurement voltage was set to *U* = 10 V and the measurement time to 10 s. After the current curve was stable, the average value was taken, and Formula (3) was used to calculate the conductivity σ of the material, the unit is S/m.
(3)$$\sigma =\frac{I}{U}\times \frac{D}{LW}$$

## 3. Results and Discussion

### 3.1. TEM

TEM is performed to observe the dispersion of PTFE and MWCNTs. Figure 2a is the TEM of a melt blending of PTFE/PP. It can be seen that the melt blending only provides shear force and is not enough to make the fiber in situ fibrillation. Figure 2b displays the TEM of PP/PTFE/0.2% MWCNTs. The black strips in the red circle are MWCNTs, with a diameter of about 10 nm and a length of about 400 nm. In comparison with the purchased MWCNTs, the length is 10–20 μm and the diameter is larger than 50 nm, which is much smaller. This may be because the MWCNTs are broken by the strong shearing and stretching action of the screw. It can also be seen that MWCNTs are not all dispersed at the two–phase interface, but a few are dispersed in the resin. Figure 2c reveals that the inside of the fiber consists of hollow microspheres, which also proves that it is a PTFE fiber. The mechanism of in situ fibrillation of PTFE has been reported before [31]. As long as the processing temperature is lower than the melting point of PTFE, the morphology of PTFE fiber can be maintained because the molecular chain cannot be relaxed and cannot return to its original shape. Figure 2d reveals the TEM of PP/PTFE/1% MWCNTs. The red dashed circle in the figure is MWCNT, which are dispersed in the phase interface of PTFE and PP. The aspect ratio is similar to Figure 2b and is consistent with the experimental concept. It is easier to form a conductive network when dispersed in the phase interface, so can achieve high conductivity under low loading.

### 3.2. SEM

Figure 3 is the SEM of PP/1% PTFE and PP/3% PTFE after etching PP with p–xylene. The diameter of fiber in Figure 3a is about 5 μm and the diameter of fiber in Figure 3b is nanoscale, which is close to the parameters of the purchased PTFE (the average particle size is 9 μm, and the primary particle size is 200 nm). It can be seen that the PTFE powder has constructed in situ fibrillation into fibers by shearing and stretching during the twin–screw extrusion process. The literature reports that in situ fibrillation of PTFE can greatly improve the foaming ability of PP by promoting crystallization, and enhancing the viscoelasticity and strength of the melt [16,17,23,32].

Figure 4 reveals PP/PTFE composites with different contents of PTFE from the injection molding foam skin layer to the core layer. It can be seen that there are obvious skin layer structures, while the cell diameter and the cell density gradually increase from the skin layer to the core layer. At the same time, the in situ fibrillation of PTFE has an obvious promotion effect on PP foaming. With the increase of PTFE contents, the cell diameter gradually decreases and the cell density gradually increases. This is because PTFE promotes the crystallization of PP and increases the melt strength.

Figure 5 displays a SEM picture of PP/PTFE composite foams with different contents of PTFE. It can be seen from that the heterogeneous nucleation efficiency of PTFE is very high. With the increase of PTFE content, the average cell diameter decreases from 17.31 μm to 7.98 μm, and the average cell density increases from 1.70 × 10^7^ units/cm^3^ to 9.93 × 10^7^ units/cm^3^. This is because PTFE promotes the crystallization of PP and improves the viscoelasticity and strength of the melt. At the same time, it can be seen from the previous TEM and SEM that the aspect ratio of PTFE in situ fibrillation is very high, which can reduce the free energy barrier of nucleation, increase the nucleation sites and heterogeneous nucleation rate, and thereby significantly improve the foaming properties of PP [33].

### 3.3. DSC

Figure 6 is the DSC curve of PP/PTFE and PP/PTFE/MWCNTs composites with different contents of PTFE and MWCNTs. Table 4 displays the thermal parameters of DSC. Crystallinity is the ratio of the enthalpy of melting to the melting enthalpy of PP at 100% crystallization (209 J·g^−1^) [34]. It can be seen from Figure 6a that with the addition of PTFE, the peak shifts to the left and the peak area increases. The crystallinity is largest when the amount of PTFE is 3%, reaching 42.6%. Meanwhile, Figure 6b plotted the DSC curve during cooling, and it can be seen that there is an increase in the polymer crystallization temperature after the addition of PTFE. These phenomena indicate that the PTFE addition increases the crystallinity and promotes the crystallization of PP. Hence, PTFE can not only be used as a nucleating agent, but also can induce PP to crystallize along the direction of PTFE fibers.

Figure 6c,d plots the DSC curves of PP/PTFE composites with different contents of MWCNTs during heating and cooling. It can be seen that with the addition of MWCNTs, the crystallization temperature and crystallinity are basically unchanged; this is due to the MWCNTs being mainly dispersed at the phase interface and having little effect on the PP phase.

### 3.4. Rheology

Figure 7a depicts the tanδ–frequency curves of pure PP, PP/PTFE and PP/PTFE/MWCNTs composites. It can be seen that pure PP exhibits typical viscoelastic fluid behavior, and tanδ decreases with increasing frequency over the entire frequency sweep range. As the content of PTFE increases, its tanδ gradually decreases. Tanδ of PP/PTFE composites increase with the increase in frequency at low frequency, and this phenomenon is a physical gel property. In the high–frequency region, the tanδ decreases with the increase in frequency, showing the same trend as PP. Due to the PTFE fiber, the network is unentangled under high shear and the composite is transformed into viscoelastic fluid behavior. Figure 7b is the G’–frequency curve of PP and PP/PTFE composites. It can be seen that, with the addition of PTFE, the storage modulus G’ increases, and G’ gradually increases with the increase of PTFE content. This is because the addition of PTFE increases the stiffness of the material, so that its energy storage modulus gradually increases. Figure 7c is the G”–frequency curve of PP and PP/PTFE composites, and it reveals that with the addition of PTFE, the loss modulus G” shows the same trend as the energy storage modulus, which is also due to the increased rigidity; the energy required for the movement of the molecular chain increases, and the loss modulus increases [15].

Figure 7d depicts the tanδ–frequency curves of PP and PP/PTFE/MWCNTs composites. It can be seen that both PP and PP/PTFE/MWCNTs exhibit typical viscoelastic fluid behavior, and tanδ decreases with increasing frequency over the entire frequency sweep range. Figure 7e,f are the G’, G”–frequency curves of PP and PP/PTFE/MWCNTs composites, respectively, it can be seen that with the addition of MWCNTs, their rigidity increases, the energy storage modulus G’ increases, and the energy required for molecular chain movement increases, thereby increasing the loss modulus G”. It also can be found that the curves of G’, G” form a gel point under the addition of 0.2 MWCNTs, which proves the construction of three–dimensional network and indirectly proves the formation of conductive network, thus consistence with the high conductivity under low loading.

### 3.5. Foam Density

The density of PP, PTFE, and MWCNTs are 0.90 g/cm^3^, 0.40–0.55 g/cm^3^, 2.20 g/cm^3^, respectively. From Table 5, we can see that with the increase in PTFE, the density of PP/PTFE gradually decreases; when the PTFE content is 5%, the density is reduced to 0.72 g/cm^3^, and after injection foaming, further weight reduction as low as 0.43 g/cm^3^, and the expansion ratio reaches 1.67. This is because PTFE in situ fibrillation, as a heterogeneous nucleation agent [19], promotes nucleation and improves the foaming ability of PP, which is consistent with SEM. After the addition of MWCNTs, the density gradually increased. After injection foaming, its density gradually decreased with the increase in MWCNTs, as low as 0.35 g/cm^3^, and the expansion ratio reached 2.46. This is due to the fact that MWCNTs are distributed at the phase interface and act as nucleating agents to induce nucleation, further increasing expansion ratio.

### 3.6. Tensile Testing

It can be seen from Figure 8a that after the addition of PTFE, the tensile strength increases, and gradually increases with the increase in content. The tensile strength increases from 13.6 MPa to 16.8 MPa when the PTFE content is 3%; his is because the addition of PTFE enhances the strength. Its elongation at break decreases slightly with the addition of PTFE, and remains at about 17%, which is relatively brittle. Figure 8b plots the tensile curve of PP/PTFE composite foams, and it can be seen that the tensile strength is significantly reduced from 15.4 MPa to 11.1 MPa after foaming. This corresponds to the cortical structure of SEM, and foamed PP has the thickest cortex thickness; it reaches 250 μm and, therefore, the highest tensile strength. The elongation at break is slightly increased. Thus PP/3% PTFE is the optimal component.

### 3.7. Conductivity Test

Figure 9 depicts the conductivity of PP/PTFE/MWCNTs composites with different contents of MWCNTs before and after foaming. It can be seen that, after the addition of 0.2% MWCNTs, the conductivity suddenly increases to 2.73 × 10^−5^ S/m, and its conductivity gradually increases with the increase in MWCNTs, reaching 3.41 × 10^−4^ S/m under the addition of 1% MWCNTs. This is because MWCNTs are dispersed between the two phases of PP and PTFE, so that the content required to form a conductive path is significantly reduced to as low as 0.2%. However, its conductivity decreases slightly after foaming, probably because the foaming process destroys fraction conductive networks, making its conductivity slightly reduced.

## 4. Conclusions

This paper used PP as the continuous phase, PTFE as the dispersed phase, and MWCNTs as the conductive filler. This study successfully prepared different contents of PP/PTFE, PP/PTFE/MWCNTs in situ fibrillation composite foams by solution blending, twin–screw extrusion and injection foaming. The effects of different PTFE contents on PP were explored, and the results revealed that the increase of PTFE content could promote PP crystallization and improve PP foaming ability. Meanwhile, PTFE with 3% content is sufficient to obtain excellent performance, so the optimal component is determined to be 3% PP/PTFE/MWCNTs in situ fibrillation composite foams were successfully prepared, most of which were uniformly distributed at the phase interface of PP/PTFE, which makes it easier to form a conductive network and achieve high conductivity under low loading.

## Figures and Tables

**Figure 1 polymers-14-05411-f001:**
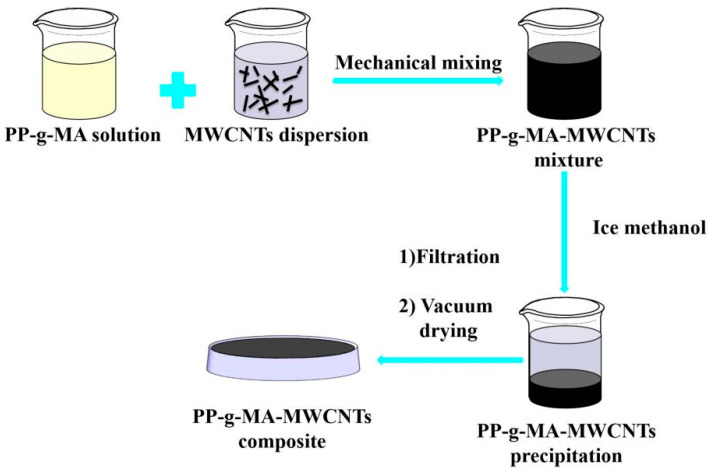
Preparation process.

**Figure 2 polymers-14-05411-f002:**
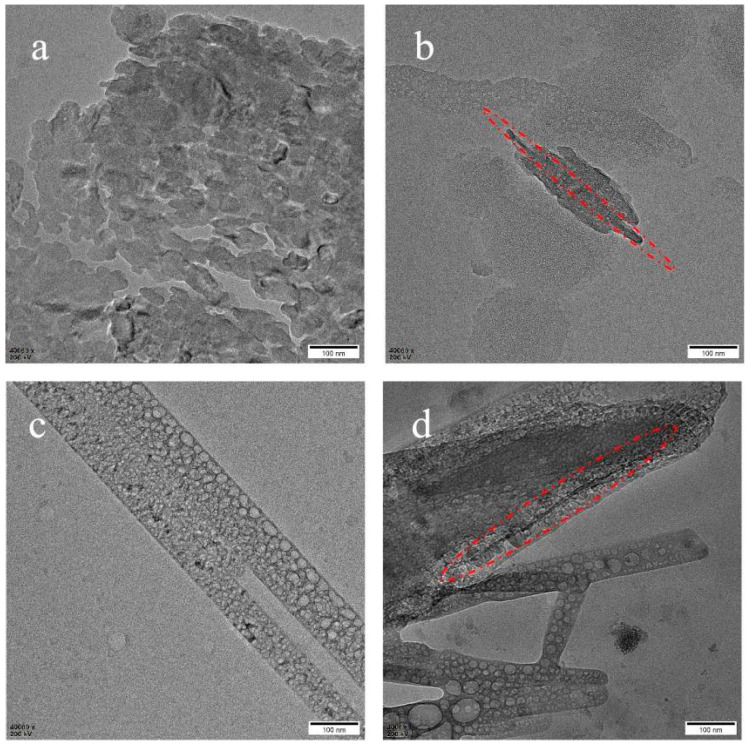
TEM of (**a**) melt blended PP/PTFE; (**b**) PP/PTFE/0.2% MWCNTs; (**c**) PP/PTFE/0.5% MWCNTs; (**d**) PP/PTFE/1% MWCNTs.

**Figure 3 polymers-14-05411-f003:**
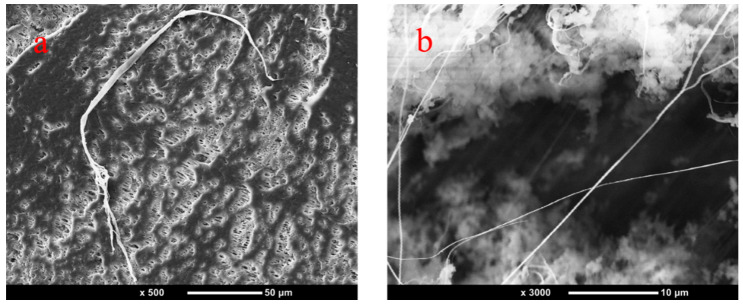
(**a**) SEM picture of PP/1% PTFE after PP etched with p–xylene, (**b**) SEM picture of PP/3% PTFE after PP etched with p–xylene.

**Figure 4 polymers-14-05411-f004:**
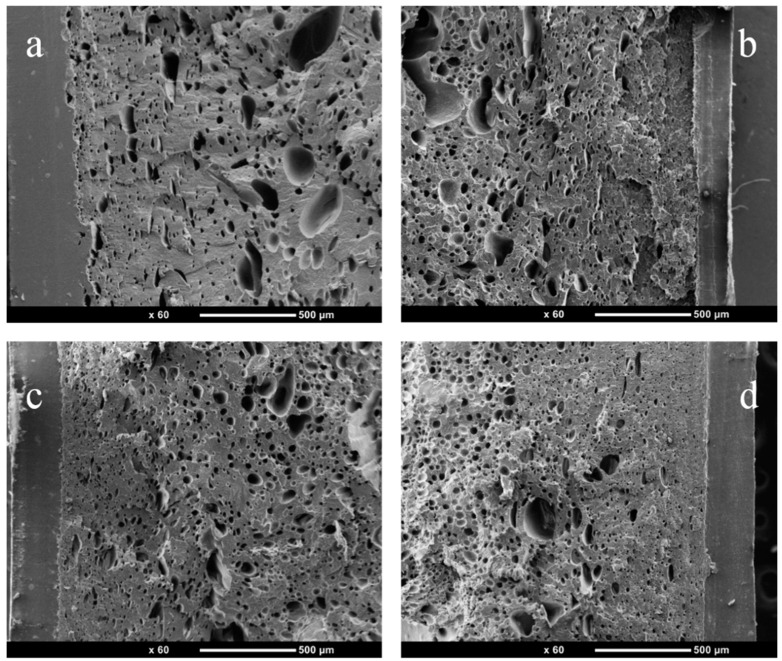
SEM picture of PP/PTFE composite materials with different contents of PTFE using foam injection molding from the skin to the core layer (**a**) PP; (**b**) 1% PTFE; (**c**) 3% PTFE; (**d**) 5% PTFE.

**Figure 5 polymers-14-05411-f005:**
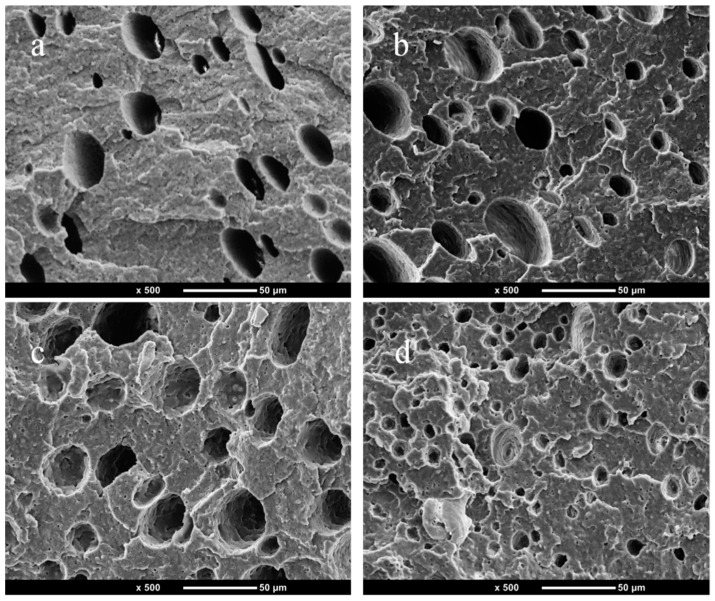
SEM picture of injection molded foam of PP/PTFE composite material with different contents of PTFE (**a**) PP; (**b**) 1% PTFE; (**c**) 3% PTFE; (**d**) 5% PTFE.

**Figure 6 polymers-14-05411-f006:**
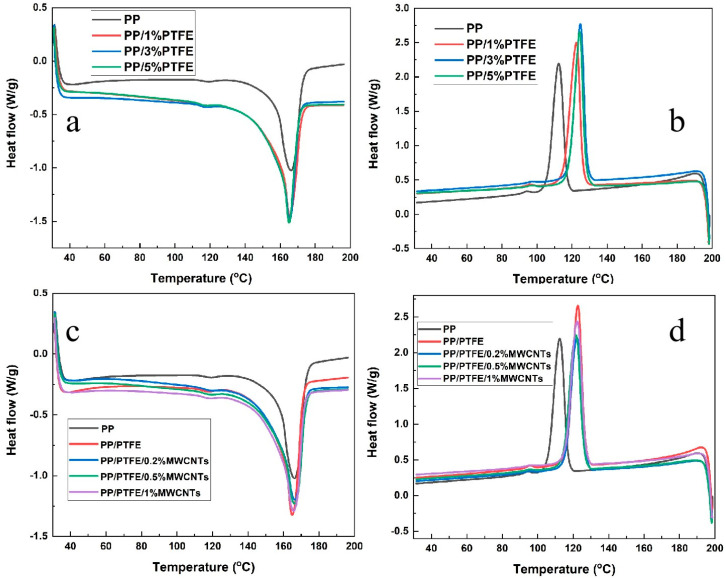
DSC curve of PP/PTFE composites with different contents of PTFE (**a**) heating; (**b**) cooling; DSC curve of PP/PTFE/MWCNTs composites with different contents of MWCNTs (**c**) heating; (**d**) cooling.

**Figure 7 polymers-14-05411-f007:**
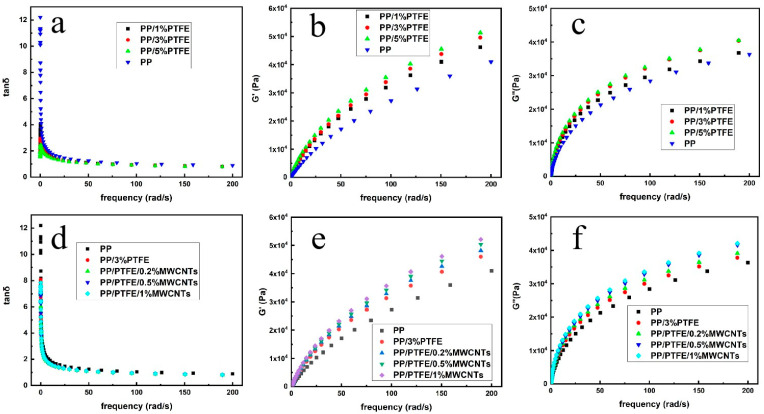
Rheological properties of PP, PP/PTFE, PP/PTFE/MWCNTs (**a**,**d**) tanδ; (**b**,**e**) G’; (**c**,**f**) G”.

**Figure 8 polymers-14-05411-f008:**
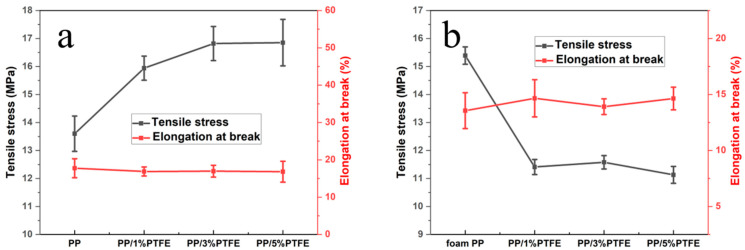
PP and PP/PTFE composite tensile curve before and after foaming (**a**) before foaming; (**b**) after foaming.

**Figure 9 polymers-14-05411-f009:**
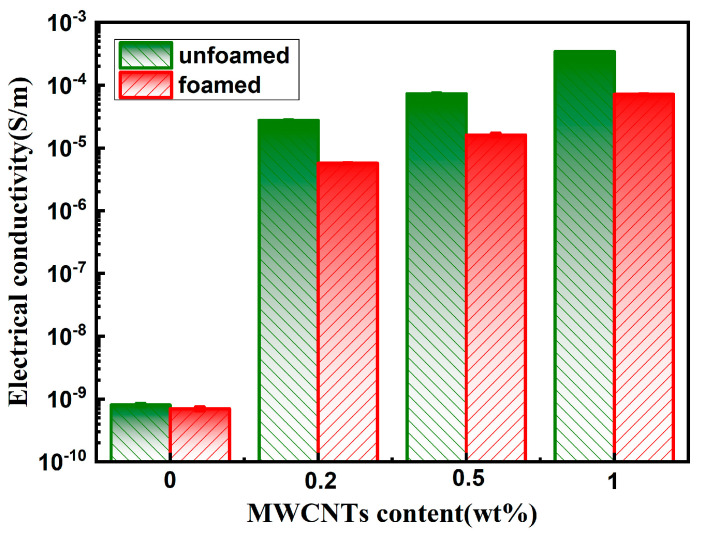
Conductivity of PP/PTFE/MWCNTs composites with different contents of MWCNTs before and after foaming.

**Table 1 polymers-14-05411-t001:** Extrusion processing temperature.

Zone	1	2	3	4	5	6	7	8
Temperature (°C)	155	155	160	170	180	180	180	180

**Table 2 polymers-14-05411-t002:** Masterbatch composition and formula ratio.

Masterbatch Composition	PP (wt%)	PP–g–MA (wt%)	MWCNTs (wt%)
PP/5% MWCNTs	85.5	9.5	5

**Table 3 polymers-14-05411-t003:** Sample formula.

Sample Formula	PP (wt%)	PTFE (wt%)	Masterbatch (wt%)
PP/PTFE	97	3	0
PP/PTFE/0.2% MWCNTs	93.58	3	4
PP/PTFE/0.5% MWCNTs	88.45	3	10
PP/PTFE/1% MWCNTs	79.9	3	20

**Table 4 polymers-14-05411-t004:** Thermal parameters of DSC.

	Crystallization Temperature (°C)	Melting Temperature (°C)	ΔH (J·g^−1^)	Crystallinity
PP	120.6	168.4	74.64	35.7%
PP/1% PTFE	122.5	165.5	87.58	41.9%
PP/3% PTFE	122.7	165.0	88.95	42.6%
PP/5% PTFE	124.3	165.1	84.35	40.4%
PP/PTFE/0.2% MWCNT	122.2	166.6	80.03	38.3%
PP/PTFE/0.5% MWCNT	121.5	166.3	83.23	39.8%
PP/PTFE/1% MWCNT	122.3	165.7	84.88	40.6%

**Table 5 polymers-14-05411-t005:** Density of Composite before and after foaming.

	Density before Foaming (g/cm^3^)	Density after Foaming (g/cm^3^)	Expansion Ratio
PP	0.90	0.73	1.23
PP/1% PTFE	0.77	0.55	1.40
PP/3% PTFE	0.75	0.50	1.50
PP/5% PTFE	0.72	0.43	1.67
PP/3% PTFE/0.2% MWCNTs	0.79	0.40	1.98
PP/3% PTFE/0.5% MWCNTs	0.82	0.38	2.16
PP/3% PTFE/1% MWCNTs	0.86	0.35	2.46

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
