# Peer review of "The Injected Foaming Study of Polypropylene/Multiwall Carbon Nanotube Composite with In Situ Fibrillation Reinforcement"

_polymers, 2022, doi:10.3390/polym14245411_

Round 1

Reviewer 2 Report

The authors have systematically explored the in-situ fibrillation process in reinforced polypropylene (PP) by dispersing multiwalled carbon nanotubes, PP grafted maleic anhydride via the foam injection molding process and its impact on mechanical and electrical conductivity.  

The conception of this work is good and the experiments are conducted systematically with appropriate discussion. This work is fit for publishing pending these minor changes:

1) In the introduction section the authors need to better elaborate on the novelty of their study and how it is different from those of Wang and Park et al. 

2) The authors need to clearly list the equipment and processing conditions that were used to prepare the masterbatch, its subsequent dispersion and foam injection molding (What ScF gas was used ? How much wt % ScF was used ? Was core back retraction used in that case please provide details of the mold, was rapid cooling used ?). 

3) Similarly what make of SEM, DSC, Rheometer and TEM were used ? What were the operating conditions ? How were the samples prepared ?

4) The authors discuss the crystallinity changes with an increase in MWCNT %. Please prepare a table listing the enthalpy of melting and the corresponding % crystallinity. 

5) Figure 9 needs error bars. 

Reviewer 3 Report

Li et al paper titled on “The injected foaming study of polypropylene/multiwall carbon nanotube composite with in-situ fibrillation reinforcement”The overall presentation of this work is good. It is one of the interesting research topics in automotive industry. Therefore, I must accept this work in journal after the following minor revision.

-Make sure all abbreviations are written out in full the first time.

-          - polymere-fibrillar would be written as polymer-fibrillar

-          -Provide space between Temperature and degree Celsius in in Table 1. Check this kind of mistake in other tables also.

-          -Author requested to cite literature evidenced to support your experimental output like TEM, SEM, DSC, Density, Conductivity in the results and discussion sections.

-          -Why does author used polytetrafluoroethylene (PTFE) as dispersed phase. Provide advantages of using PTFE over others dispersed phase.

Reviewer 4 Report

Dear,

The manuscript developed conductive polymeric foams. The manuscript needs a strong review in the discussion of the results, as well as citations and comparison with the literature. Furthermore, the authors do not correlate the properties with each other. Here are some additional suggestions:

> Abstract. Please better present the results in the abstract, especially the most promising formulation;

> Please correct the term PP-G-M to PP-g-MA

> Page 1. “In recent years, with the development of..................”. Please add references in this paragraph;

> The introduction needs a broader discussion about the novelty and the gap on the topic;

> Materials. Informs density and melt flow rate for PTFE and PP-g-MA. In addition, inform the degree of maleic anhydride grafting of PP-g-MA;

> Page 2. “and then add them to the feed port of the extruder for extrusion to obtain a PP/PTFE composites....”. Inform the feed rate in the extruder;

> Rheology test. Why did the authors only use this range of 0.03 – 200 rad/s? In general, studies are conducted up to 600 rad/s;

> Page 8. DSC. Discuss in detail the thermal properties: crystalline melting temperature; crystallization temperature and degree of crystallinity. I recommend adding a table with all the thermal parameters. Also, calculate the degree of crystallinity;

> Page 9. “PTFE can not only be used as a nucleating agent, but also can induce PP to crystallize along the direction of PTFE fibers”. Please add the curve with the crystallization temperature, so you can validate the claim;

> Rheology. The authors need to deepen the discussion, for example, relate it to the electrical percolation threshold;

> The manuscript lacks literature data to inform application;

Round 2

Reviewer 4 Report

Dear, 

The authors improved the quality of the manuscript, generating greater clarity. Therefore, the manuscript has merit for publication. 

Yours sincerely,